# DiskANN: Fast Accurate Billion-point Nearest Neighbor Search on a Single Node

**Suhas Jayaram Subramanya**[*]
Carnegie Mellon University
suhas@cmu.edu

**Devvrit**[*]
University of Texas at Austin
devvrit.03@gmail.com

**Rohan Kadekodi**[*]
University of Texas at Austin
rak@cs.texas.edu

**Ravishankar Krishaswamy**
Microsoft Research India
rakri@microsoft.com

**Harsha Vardhan Simhadri**
Microsoft Research India
harshasi@microsoft.com

## Abstract

Current state-of-the-art approximate nearest neighbor search (ANNS) algorithms generate indices that must be stored in main memory for fast high-recall search. This makes them expensive and limits the size of the dataset. We present a new graph-based indexing and search system called DiskANN that can index, store, and search a billion point database on a single workstation with just 64GB RAM and an inexpensive solid-state drive (SSD). Contrary to current wisdom, we demonstrate that the SSD-based indices built by DiskANN can meet all three desiderata for large-scale ANNS: high-recall, low query latency and high density (points indexed per node). On the billion point SIFT1B *bigann* dataset, DiskANN serves $> 5000$ queries a second with $< 3$ms mean latency and $95\%+$ 1-recall@1 on a 16 core machine, where state-of-the-art billion-point ANNS algorithms with similar memory footprint like FAISS [18] and IVFOADC+G+P [8] plateau at around $50\%$ 1-recall@1. Alternately, in the high recall regime, DiskANN can index and serve $5 - 10$x more points per node compared to state-of-the-art graph-based methods such as HNSW [21] and NSG [13]. Finally, as part of our overall DiskANN system, we introduce Vamana, a new graph-based ANNS index that is more versatile than the existing graph indices even for in-memory indices.

## 1   Introduction

In the nearest neighbor search problem, we are given a dataset $P$ of points in some space. The goal is to design a data structure of small size, such that, for any query $q$ in the same metric space, and target $k$, we can retrieve the $k$ nearest neighbors of $q$ from the dataset $P$ quickly. This is a fundamental problem in algorithms research, and also a commonly used sub-routine in a diverse set of areas such as computer vision, document retrieval and recommendation systems, to name a few. In these applications, the actual entities — images, documents, user profiles — are *embedded* into a hundred or thousand dimensional space such that a desired notion of the entities' similarity is encoded as distance between their embeddings.

Unfortunately, it is often impossible to retrieve the exact nearest neighbors without essentially resorting to a linear scan of the data (see, e.g., [15, 23]) due to a phenomenon known as the *curse of dimensionality* [10]. As a result, one resorts to finding the *approximate nearest neighbors* (ANN) where the goal is to retrieve $k$ neighbors which are close to being optimal. More formally, consider a query $q$, and suppose the algorithm outputs a set $X$ of $k$ candidate near neighbors, and suppose $G$ is

---

[*]Work done while at Microsoft Research India

the ground-truth set of the $k$ closest neighbors to $q$ from among the points of the base dataset. Then, we define the $k$-recall@$k$ of this set $X$ to be $\frac{|X \cap G|}{k}$. The goal of an ANN algorithm then is to maximize recall while retrieving the results as quickly as possible, which results in the recall-vs-latency tradeoff.

There are numerous algorithms for this problem with diverse index construction methodologies and a range of tradeoffs w.r.t indexing time, recall, and query time. For example, while k-d trees generate compact indices that are fast to search in low dimensions, they are typically very slow when dimension $d$ exceeds about 20. On the other hand, Locality Sensitive Hashing based methods [2, 4] provide *near-optimal* guarantees on the tradeoff between index size and search time in the worst case, but they fail to exploit the distribution of the points and are outperformed by more recent graph-based methods on real-world datasets. Recent work on data-dependent LSH schemes (e.g. [3]) is yet to be proven at scale. As of this writing, the best algorithms in terms of search time vs recall on real-world datasets are often graph-based algorithms such as HNSW [21] and NSG [13] where the indexing algorithm constructs a *navigable* graph over the base points, and the search procedure is a best-first traversal that starts at a chosen (or random) point, and walks along the edges of the graph, while getting closer to the query at each step until it converges to a local minimum. A recent work of Li et al. [20] has an excellent survey and comparison of ANN algorithms.

Many applications require fast and accurate search on billions of points in Euclidean metrics. Today, there are essentially two high-level approaches to indexing large datasets.

The first approach is based on *Inverted Index + Data Compression* and includes methods such as FAISS [18] and IVFOADC+G+P [8]. These methods cluster the dataset into $M$ partitions, and compare the query to only the points in a few, say, $m \ll M$ partitions closest to the query. Moreover, since the full-precision vectors cannot fit in main memory, the points are compressed using a quantization scheme such as *Product Quantization* [17]. While these schemes have a small memory footprint – less than 64 GB for storing an index on billion points in 128 dimensions – and can retrieve results in $< 5$ ms using GPUs or other hardware accelerators, their 1-recall@1 is rather low (around 0.5) since the data compression is lossy. These methods report higher recall values for a weaker notion of 1-recall@100 – the likelihood that the true nearest neighbor is present in a list of 100 output candidates. However, this measure may not be acceptable in many applications.

The second approach is to divide the dataset into disjoint *shards*, and build an in-memory index for each shard. However, since these indices store both the index and the uncompressed data points, they have a larger memory footprint than the first approach. For example, an NSG index for 100M floating-point vectors in 128 dimensions would have a memory footprint of around 75GB[2]. Therefore, serving an index over a billion points would need several machines to host the indices. Such a scheme is reportedly [13] in use in Taobao, Alibaba's e-commerce platform, where they divide their dataset with 2 billion 128-dimensional points into 32 shards, and host the index for each shard on a different machine. Queries are routed to all shards, and the results from all shards are aggregated. Using this approach, they report 100-recall@100 values of 0.98 with a latency of $\sim 5$ms. Note that extending this to web scale data with *hundreds of billions* of points would require thousands of machines.

The scalability of both these classes of algorithms is limited by the fact that they construct indices meant to be served from main memory. Moving these indices to disks, even SSDs, would result in a catastrophic rise of search latency and a corresponding drop in throughput. The current wisdom on search requiring main memory is reflected in the blog post by FAISS [11]: *"Faiss supports searching only from RAM, as disk databases are orders of magnitude slower. Yes, even with SSDs."*

Indeed, the search throughput of an SSD-resident index is limited by the number of random disk accesses/query and latency is limited by the the number of round-trips (each round-trip can consist of multiple reads) to the disk. An inexpensive retail-grade SSD requires a few hundred microseconds to serve a random read and can service about $\sim 300$K random reads per second. On the other hand, search applications (e.g. web search) with multi-stage pipelines require mean latencies of a few milliseconds for nearest neighbor search. Therefore, the main challenges in designing a performant SSD-resident index lie in reducing (a) the number of random SSD accesses to a few dozen, and (b) the number of round trip requests to disk to under ten, preferably five. Naively mapping indices generated by traditional in-memory ANNS algorithms to SSDs would generate several hundreds of disk reads per query, which would result in unacceptable latencies.

## 1.1 Our technical contribution

We present DiskANN, an SSD-resident ANNS system based on our new graph-based indexing algorithm called Vamana, that debunks current wisdom and establishes that even commodity SSDs can effectively support large-scale ANNS. Some interesting aspects of our work are:

- DiskANN can index and serve a billion point dataset in 100s of dimensions on a workstation with 64GB RAM, providing 95%+ 1-recall@1 with latencies of under 5 milliseconds.
- A new algorithm called Vamana which can generate graph indices with smaller diameter than NSG and HNSW, allowing DiskANN to minimize the number of sequential disk reads.
- The graphs generated by Vamana can be also be used in-memory, where their search performance matches or exceeds state-of-the-art in-memory algorithms such as HNSW and NSG.
- Smaller Vamana indices for overlapping partitions of a large dataset can be easily merged into one index that provides nearly the same search performance as a single-shot index constructed for the entire dataset. This allows indexing of datasets that are otherwise too large to fit in memory.
- We show that Vamana can be combined with off-the-shelf vector compression schemes such as product quantization to build the DiskANN system. The graph index along with the full-precision vectors of the dataset are stored on the disk, while compressed vectors are cached in memory.

## 1.2 Notation

For the remainder of the paper, we let $P$ denote the dataset with $|P| = n$. We consider directed graphs with vertices corresponding to points in $P$, and edges between them. With slight notation overload, we refer to such graphs as $G = (P, E)$ by letting $P$ also denote the vertex set. Given a point $p \in P$ in a directed graph, we let $N_{\text{out}}(p)$ to denote the set of out-edges incident on $p$. Finally, we let $\mathsf{x}_p$ denote the vector data corresponding to $p$, and let $d(p, q) = ||\mathsf{x}_p - \mathsf{x}_q||$ denote the metric distance between two points $p$ and $q$. All experiments presented in this paper used Euclidean metric.

## 1.3 Paper Outline

Section 2 presents Vamana our new graph index construction algorithm and Section 3 explains the overall system design of DiskANN. Section 4 presents an empirical comparison Vamana with HNSW and NSG for in-memory indices, and also demonstrates the search characteristics of DiskANN for billion point datasets on a commodity machine.

## 2 The Vamana Graph Construction Algorithm

We begin with a brief overview of graph-based ANNS algorithms before presenting the details of Vamana, a specification which is given in Algorithm 3.

### 2.1 Relative Neighborhood Graphs and the GreedySearch algorithm

Most graph-based ANNS algorithms work in the following manner: during index construction, they build a graph $G = (P, E)$ based on the geometric properties of the dataset $P$. At search time, for a query vector $\mathsf{x}_q$, search employs a natural greedy or best-first traversal, such as in Algorithm 1, on $G$. Starting at some designated point $s \in P$, they traverse the graph to get progressively closer to $\mathsf{x}_q$.

There has been much work on understanding how to construct sparse graphs for which the GreedySearch$(s, \mathsf{x}_q, k, L)$ converges quickly to the (approximate) nearest neighbors for any query. A sufficient condition for this to happen, at least when the queries are close to the dataset points, is the so-called *sparse neighborhood graph* (SNG), which is introduced in [5][3]. In an SNG, the out-neighbors of each point $p$ are determined as follows: initialize a set $S = P \setminus \{p\}$. As long as $S \neq \emptyset$, add a directed edge from $p$ to $p^*$, where $p^*$ is the closest point to $p$ from $S$, and remove from $S$ all points $p'$ such that $d(p, p') > d(p^*, p')$. It is then easy to see that GreedySearch$(s, \mathsf{x}_p, 1, 1)$ starting at any $s \in P$ would converge to $p$ for all base points $p \in P$.

While this construction is ideal in principle, it is infeasible to construct such graphs for even moderately sized datasets, as the running time is $\widetilde{O}(n^2)$. Building on this intuition, there have been a series of works that design more practical algorithms that generate good approximations of SNGs [21, 13]. However, since they all essentially try to approximate the SNG property, there is very little flexibility in controlling the diameter and the density of the graphs output by these algorithms.

**Algorithm 1:** GreedySearch($s, \mathsf{x}_q, k, L$)

**Data:** Graph $G$ with start node $s$, query $\mathsf{x}_q$, result size $k$, search list size $L \geq k$

**Result:** Result set $\mathcal{L}$ containing $k$-approx NNs, and a set $\mathcal{V}$ containing all the visited nodes

**begin**

    `initialize sets` $\mathcal{L} \leftarrow \{s\}$ `and` $\mathcal{V} \leftarrow \emptyset$

    **while** $\mathcal{L} \setminus \mathcal{V} \neq \emptyset$ **do**

        `let` $p* \leftarrow \arg\min_{p \in \mathcal{L} \setminus \mathcal{V}} \|\mathsf{x}_p - \mathsf{x}_q\|$

        `update` $\mathcal{L} \leftarrow \mathcal{L} \cup N_{\text{out}}(p^*)$ `and`
$\mathcal{V} \leftarrow \mathcal{V} \cup \{p^*\}$

        **if** $|\mathcal{L}| > L$ **then**

            `update` $\mathcal{L}$ `to retain closest` $L$ `points to` $\mathsf{x}_q$

    `return` [`closest` $k$ `points from` $\mathcal{L}$; $\mathcal{V}$]

---

**Algorithm 2:** RobustPrune($p, \mathcal{V}, \alpha, R$)

**Data:** Graph $G$, point $p \in P$, candidate set $\mathcal{V}$, distance threshold $\alpha \geq 1$, degree bound $R$

**Result:** $G$ is modified by setting at most $R$ new out-neighbors for $p$

**begin**

    $\mathcal{V} \leftarrow (\mathcal{V} \cup N_{\text{out}}(p)) \setminus \{p\}$

    $N_{\text{out}}(p) \leftarrow \emptyset$

    **while** $\mathcal{V} \neq \emptyset$ **do**

        $p^* \leftarrow \arg\min_{p' \in \mathcal{V}} d(p, p')$

        $N_{\text{out}}(p) \leftarrow N_{\text{out}}(p) \cup \{p^*\}$

        **if** $|N_{\text{out}}(p)| = R$ **then**

            `break`

        **for** $p' \in \mathcal{V}$ **do**

            **if** $\alpha \cdot d(p^*, p') \leq d(p, p')$ **then**

                `remove` $p'$ `from` $\mathcal{V}$

## 2.2 The Robust Pruning Procedure

As mentioned earlier, graphs which satisfy the SNG property are all good candidates for the GreedySearch search procedure. However, it is possible that the diameter of the graphs can be quite large. For example, if the points are linearly arranged on the real line in one dimension, the $O(n)$ diamater line graph, where each point connects to its two neighbors (one at the end), is the one that satisfies the SNG property. Searching such graphs stored in disks would incur many sequential reads to the disk at to fetch the neighbors of the vertices visited on the search path in Algorithm 1.

To overcome this, we would like to ensure that the distance to the query decreases by a multiplicative factor of $\alpha > 1$ at every node along the search path, instead of merely decreasing as in the SNG property. Consider the directed graph where the out-neighbors of every point $p$ are determined by the RobustPrune($p, \mathcal{V}, \alpha, R$) procedure in Algorithm 2. Note that if the out-neighbors of every $p \in P$ are determined by RobustPrune($p, P \setminus \{p\}, \alpha, n-1$), then GreedySearch($s, p, 1, 1$), starting at any $s$, would converge to $p \in P$ in logarithmically many steps, if $\alpha > 1$. However, this would result in a running time of $\widetilde{O}(n^2)$ for index construction. Hence, building on the ideas of [21, 13], Vamana invokes RobustPrune($p, \mathcal{V}, \alpha, R$) for a carefully selected $\mathcal{V}$ with far fewer than $n-1$ nodes, to improve index construction time.

## 2.3 The Vamana Indexing Algorithm

Vamana constructs a directed graph $G$ in an iterative manner. The graph $G$ is initialized so that each vertex has $R$ randomly chosen out-neighbors. Note that while the graph is well connected when $R > \log n$, random connections do not ensure convergence of the GreedySearch algorithm to good results. Next, we let $s$ denote the medoid of the dataset $P$, which will be the starting node for the search algorithm. The algorithm then iterates through all the points in $p \in P$ in a random order, and in each step, updates the graph to make it more suitable for GreedySearch($s, \mathsf{x}_p, 1, L$) to converge to $p$. Indeed, in the iteration corresponding to point $p$, Vamana first runs GreedySearch($s, \mathsf{x}_p, 1, L$) on the current graph $G$, and sets $\mathcal{V}_p$ to the *set of all points visited* by GreedySearch($s, \mathsf{x}_p, 1, L$). Then, the algorithm updates $G$ by running RobustPrune($p, \mathcal{V}_p, \alpha, R$) to determine $p$'s new out-neighbors. Then, Vamana updates the graph $G$ by adding backward edges $(p', p)$ for all $p' \in N_{\text{out}}(p)$. This ensures that there are connections between the vertices visited on the search path and $p$, thereby ensuring that the updated graph will be better suited for GreedySearch($s, \mathsf{x}_p, 1, L$) to converge to $p$.

However, adding backward edges of the form $(p', p)$ might lead to a degree violation of $p'$, and so whenever any vertex $p'$ has an out-degree which exceeds the degree threshold of $R$, the graph is modified by running RobustPrune($p', N_{\text{out}}(p'), \alpha, R$) where $N_{\text{out}}(p')$ is the set of existing out-neighbors of $p'$. As the algorithm proceeds, the graph becomes consistently better and faster for GreedySearch. Our overall algorithm makes two passes over the dataset, the first pass with $\alpha = 1$, and the second with a user-defined $\alpha \geq 1$. We observed that a second pass results in better graphs, and that running both passes with the user-defined $\alpha$ makes the indexing algorithm slower as the first pass computes a graph with higher average degree which takes longer.

**Algorithm 3:** Vamana Indexing algorithm

---

**Data:** Database $P$ with n points where $i$-th point has coords $\mathsf{x}_i$, parameters $\alpha$, L, R
**Result:** Directed graph G over $P$ with out-degree <=R
**begin**
   initialize $G$ to a random R-regular directed graph
   let s denote the medoid of dataset $P$
   let $\sigma$ denote a random permutation of $1..n$
   **for** $1 \leq i \leq n$ **do**
      let $[\mathcal{L}; \mathcal{V}] \leftarrow$ GreedySearch$(s, \mathsf{x}_{\sigma(i)}, 1, L)$
      run RobustPrune$(\sigma(i), \mathcal{V}, \alpha, \mathtt{R})$ to update out-neighbors of $\sigma(i)$
      **for** *all points* j *in* $N_{\text{out}}(\sigma(i))$ **do**
         **if** $|N_{\text{out}}(j) \cup \{\sigma(i)\}| > \mathtt{R}$ **then**
            run RobustPrune$(j, N_{\text{out}}(j) \cup \{\sigma(i)\}, \alpha, \mathtt{R})$ to update out-neighbors of $j$
         **else**
            update $N_{\text{out}}(j) \leftarrow N_{\text{out}}(j) \cup \sigma(i)$

---

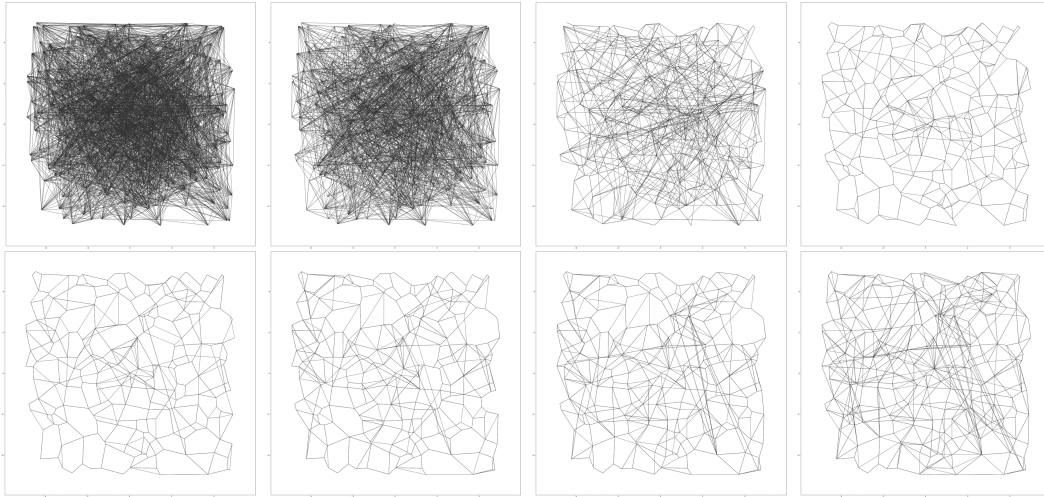

Figure 1: Progression of the graph generated by the Vamana indexing algorithm described in Algorithm 3 on a database with 200 points in 2 dimensions. Notice that the algorithm goes through the first pass with $\alpha = 1$, followed by the second pass where it introduces long range edges.

## 2.4 Comparison of Vamana with HNSW [21] and NSG [13]

At a high level, Vamana is rather similar to HNSW and NSG, two very popular ANNS algorithms. All three algorithms iterate over the dataset $P$, and use the results of the GreedySearch$(s, \mathsf{x}_p, 1, L)$ and RobustPrune$(p, \mathcal{V}, \alpha, R)$ to determine $p$'s neighbors. However, there are some important differences between these algorithms. Most crucially, both HNSW and NSG have no tunable parameter $\alpha$ and implicitly use $\alpha = 1$. This is the main factor which lets Vamana achieve a better trade-off between graph degree and diameter. Next, while HNSW sets the candidate set $\mathcal{V}$ for the pruning procedure to be the *final result-set of $\mathcal{L}$ candidates* output by GreedySearch$(s, p, 1, L)$, Vamana and NSG let $\mathcal{V}$ be the entire set of vertices visited by GreedySearch$(s, p, 1, L)$. Intuitively, this feature helps Vamana and NSG add long-range edges, while HNSW, by virtue of adding only local edges to nearby points, has an additional step of constructing a hierarchy of graphs over a nested sequence of samples of the dataset. The next difference pertains to the initial graph: while NSG sets the starting graph to be an approximate $K$-nearest neighbor graph over the dataset, which is a time and memory intensive step, HNSW and Vamana have simpler initializations, with the former beginning with an empty graph and Vamana beginning with a random graph. We have observed that starting with a random graph results in better quality graphs than beginning with the empty graph. Finally, Vamana makes two passes over the dataset, whereas both HNSW and NSG make only one pass, motivated by our observation that the second pass improves the graph quality.

# 3 DiskANN: Constructing SSD-Resident Indices

We now present the design of the DiskANN overall in two parts. In the first part, we explain the index construction algorithm, and in the second part, we explain the search algorithm.

## 3.1 The DiskANN Index Design

The high-level idea is simple: run Vamana on a dataset $P$ and store the resulting graph on an SSD. At search time, whenever Algorithm 1 requires the out-neighbors of a point $p$, we simply fetch this information from the SSD. However, note that *just storing the vector data for a billion points in* 100 *dimensions would far exceed the RAM on a workstation!* This raises two questions: how do we build a graph over a billion points, and how do we do distance comparisons between the query point and points in our candidate list at search time in Algorithm 1, if we cannot even store the vector data?

One way to address the first question would be to partition the data into *multiple smaller shards* using clustering techniques like $k$-means, build a separate index for each shard, and route the query only to a few shards at search time. However, such an approach would suffer from increased search latency and reduced throughput since the query needs to be routed to several shards.

Our idea is simple in hindsight: *instead of routing the query to multiple shards at search time, what if we send each base point to multiple nearby centers to obtain overlapping clusters?* Indeed, we first partition a billion point dataset into $k$ clusters (with $k = 40$, say) using $k$-means, and then assign each base point to the $\ell$-*closest centers* (typically $\ell = 2$ suffices). We then build Vamana indices for the points assigned to each of the clusters (which would now only have about $N\frac{\ell}{k}$ points and thus can be indexed in-memory), and finally merge all the different graphs into a single graph by taking a simple union of edges. Empirically, it turns out that the overlapping nature of the different clusters provides sufficient connectivity for the GreedySearch algorithm to succeed even if the query's nearest neighbors are actually split between multiple shards. We would like to remark that there have been earlier works [9, 22] which construct indices for large datasets by merging several smaller, overlapping indices. However, their ideas for constructing the overlapping clusters are different, and a more detailed comparison of these different techniques needs to be done.

Our next and natural idea to address the second question is to store *compressed vectors* $\widetilde{x}_p$ for every database point $p \in P$ in main memory, along with storing the graph on the SSD. We use a popular compression scheme known as *Product Quantization*[17][4], which encodes the data and query points into short *codes* (e.g., 32 bytes per data point) that can be used to efficiently obtain approximate distances $d(\widetilde{x}_p, x_q)$ at query time in Algorithm 1. We would like to remark that Vamana uses *full-precision coordinates* when building the graph index, and hence is able to efficiently guide the search towards the right region of the graph, although we use only the compressed data at search time.

## 3.2 DiskANN Index Layout

We store the compressed vectors of all the data points in memory, and store the graph along with the full-precision vectors on the SSD. On the disk, for each point $i$, we store its full precision vector $x_i$ followed by the identities of its $\leq R$ neighbors. If the degree of a node is smaller than $R$, we pad with zeros, so that computing the offset within the disk of the data corresponding to any point $i$ is a simple calculation, and does not require storing the offsets in memory. We will explain the need to store full-precision coordinates in the following section.

## 3.3 DiskANN Beam Search

A natural way to search for neighbors of a given query $x_q$ would be to run Algorithm 1, fetching the neighborhood information $N_{\text{out}}(p^*)$ from the SSD as needed. Distance calculations to guide the best vertices (and neighborhoods) to read from disk can be done using the compressed vectors. While reasonable, this requires many rountrips to SSD (which take few hundred microseconds) resulting in higher latencies. To reduce the number of round triprs to SSD (to fetch neighborhoods sequentially) without increasing compute (distance calculations) excessively, we fetch the neighborhoods of a small number, $W$ (say $4, 8$), of the closest points in $\mathcal{L} \setminus \mathcal{V}$ in one shot, and update $\mathcal{L}$ to be the top $L$ candidates in $\mathcal{L}$ along with all the neighbors retrieved in this step. Note that fetching a small number of random sectors from an SSD takes almost the same time as one sector. We refer to this modified search algorithm as BeamSearch. If $W = 1$, this search resembles normal greedy search. Note that if $W$ is too large, say 16 or more, then both compute and SSD bandwidth could be wasted.

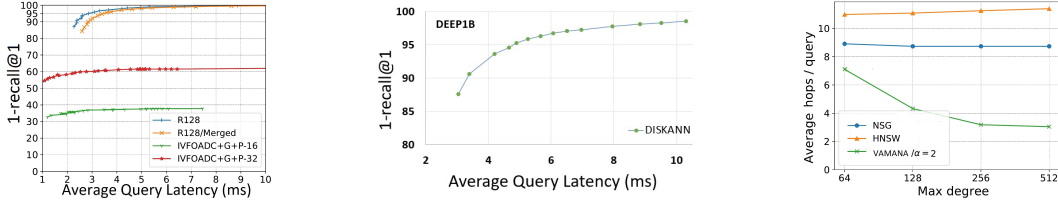

Figure 2: (a)1-recall@1 vs latency on SIFT bigann dataset. The R128 and R128/Merged series represent the one-shot and merged Vamana index constructions, respectively. (b)1-recall@1 vs latency on DEEP1B dataset. (c) Average number of hops vs maximum graph degree for achieving 98% 5-recall@5 on SIFT1M.

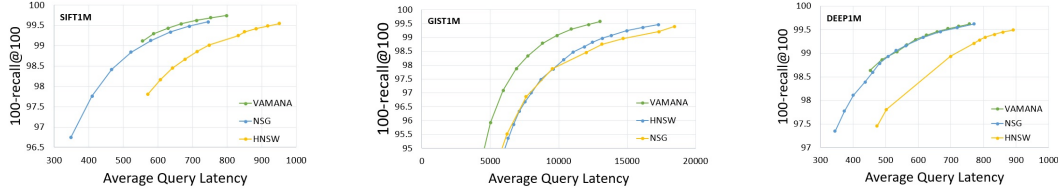

Figure 3: Latency (microseconds) vs recall plots comparing HNSW, NSG and Vamana.

Although NAND-flash based SSDs can serve 500K+ random reads per second, extracting mimaxmum read throughput requires saturating all I/O request queues. However, operating at peak throughput with backlogged queues results in disk read latencies of over a millisecond. Therefore, it is necessary to operate the SSD at a lower load factor to obtain low search latency. We have found that operating at low beam widths (e.g., $W = 2, 4, 8$) can strike a good balance between latency and throughput. In this setting, the load factor on the SSD is between $30 - 40\%$ and each thread running our search algorithm spends between $40 - 50\%$ of the query processing time in I/O.

### 3.4  DiskANN **Caching Frequently Visited Vertices**

To further reduce the number of disk accesses per query, we cache the data associated with a subset of vertices in DRAM, either based on a known query distribution, or simply by caching all vertices that are $C = 3$ or $4$ hops from the starting point $s$. Since the number of nodes in the index graph at distance $C$ grows exponentially with $C$, larger values of $C$ incur excessively large memory footprint.

### 3.5  DiskANN **Implicit Re-Ranking Using Full-Precision Vectors**

Since Product Quantization is a lossy compression method, there is a discrepancy between the closest $k$ candidates to the query computed using PQ-based approximate distances and using the actual distances. To bridge this gap, we use *full-precision coordinates* stored for each point next to its neighborhood on the disk. In fact, when we retrieve the neighborhood of a point during search, we also retrieve the full coordinates of the point without incurring extra disk reads. This is because, reading $4KB$-aligned disk address into memory is no more expensive than reading $512B$, and the neighborhood of a vertex ($4 * 128$ bytes long for degree 128 graphs) and full-precision coordinates can be stored on the same disk sector. Hence, as BeamSearch loads neighborhoods of the search frontier, it can also cache full-precision coordinates of all the nodes visited during the search process, using no extra reads to the SSD. This allows us to return the top $k$ candidates based on the full precision vectors. Independent of our work, the idea of fetching and re-ranking full-precision coordinates stored on the SSD is also used in [24], but the algorithm fetches all the vectors to re-rank in one shot, which would result in hundreds of random disk accesses all in one shot, in turn adversely affecting throughput and latency. We provide a more detailed explanation in Section 4.3. In our case, full precision coordinates *essentially piggyback* on the cost of expanding the neighborhoods.

## 4  Evaluation

We now compare Vamana with other relevant algorithms for approximate nearest neighbor search. First, for in-memory search, we compare our algorithm with NSG [13] and HNSW [21], which offer best-in-class latency vs recall on most public benchmark datasets. Next, for large billion point datasets, we compare DiskANN with compression based techniques such as FAISS [18] and IVF-OADC+G+P [8].

We use the following two machines for all experiments.

- `z840`: a bare-metal mid-range workstation with dual Xeon E5-2620v4s (16 cores), 64GB DDR4 RAM, and two Samsung 960 EVO 1TB SSDs in RAID-0 configuration.
- `M64-32ms`: a virtual machine with dual Xeon E7-8890v3s (32-vCPUs) with 1792GB DDR3 RAM that we use to build a one-shot in-memory index for billion point datasets.

## 4.1 Comparison of HNSW, NSG and Vamana **for In-Memory Search Performance**

We compared Vamana with HNSW and NSG on three commonly used public benchmarks: SIFT1M (128-dimensions) and GIST1M (960-dimensions), both of which are million point datasets of image descriptors [1], and DEEP1M (96-dimensions), a random one million size sample of DEEP1B, a machine-learned set of one billion vectors [6]. For all three algorithms, we did a parameter sweep and selected near-optimal choice of parameters for the best recall vs latency trade-off. All HNSW indices were constructed using $M = 128, ef_C = 512$, while Vamana indices used $L = 125, R = 70, C = 3000, \alpha = 2$. For NSG on SIFT1M and GIST1M, we use the parameters listed on their repository[5] due to their excellent performance, and used $R = 60, L = 70, C = 500$ for DEEP1M. Moreover, since the main focus of this work is on the SSD-based search, we did not implement our own in-mmeory search algorithm to test Vamana. Instead, we simply used the implementation of the optimized search algorithm on the NSG repository, on the indices generated by Vamana. From Figure 3, we can see one clear trend – NSG and Vamana out-perform HNSW in all instances, and on the 960-dimensional GIST1M dataset, Vamana outperforms both NSG and HNSW. Moreover, the indexing time of Vamana was better than both HNSW and NSG in all three experiments. For example, when indexing DEEP1M on `z840`, the total index construction times were 149s, 219s, and 480s for Vamana, HNSW and NSG[6] respectively. From these experiments, we conclude that Vamana matches or outperforms, the current best ANNS methods on both hundred and thousand-dimensional datasets obtained from different sources.

## 4.2 Comparison of HNSW, NSG and Vamana **for Number of Hops**

Vamana is more suitable for SSD-based serving than other graph-based algorithms as it makes $2 - 3$ times fewer hops for search to converge on large datasets compared to HNSW and NSG. By hops, we refer to the number of rounds of disk reads on the critical path of the search. In BeamSearch, it maps to the number of times the search frontier is expanded by making $W$ parallel disk reads. The number of hops is important as it directly affects search latency. For HNSW, we assume nodes in all levels excluding the base level are cached in DRAM and only count the number of hops on base-level graph. For NSG and Vamana indices, we assume that the first 3 BFS levels around the navigating node(s) can be cached in DRAM. We compare the number of hops required to achieve a target 5-recall@5 of $98\%$ by varying the maximum graph degrees in Figure 2(c), and using the BeamSearch algorithm with beamwidth of $W = 4$ for all three algorithms. We notice a stagnation trend for both HNSW and NSG, while Vamana shows a reduction in number of hops with increasing max degree, due to its ability to add more long-range edges. We thus infer that Vamana with $\alpha > 1$ makes better use of the high capacity offered by SSDs than HNSW and NSG.

## 4.3 Comparison on Billion-Scale Datasets: One-Shot Vamana **vs Merged** Vamana

For our next set of experiments, we focus our evaluations on the $10^9$ point ANN_SIFT1B [1] *bigann* dataset of SIFT image descriptors of 128 `uint8`s. To demonstrate the effectiveness of the merged-Vamana scheme described in Section 3, we built two indices using our Vamana. The first is a **single index** with $L = 125, R = 128, \alpha = 2$ on the full billion-point dataset. This procedure takes about 2 days on `M64-32ms` with a peak memory usage at $\approx$1100GB, and generates an index with an average degree of 113.9. The second is the **merged** index, which is constructed as follows: (1) partition the dataset into $k = 40$ shards using k-means clustering, (2) send each point in the dataset to the $\ell = 2$ closest shards, (3) build indices for each shard with $L = 125, R = 64, \alpha = 2$, and (4) merge the edge sets of all the graphs. The result is a $348$GB index with an average degree of $92.1$. The indices were built on `z840` and took about 5 days with memory usage remaining under 64GB for the entire process. Partitioning the dataset and merging the graphs are fast and can be done directly from the disk, and hence, the entire build process consumes under 64GB main memory.

We compare 1-recall@1 vs latency with the 10,000 query *bigann* dataset for both configurations in Figure 2(a) by running the search using 16 threads (each query is processed only on a single thread). From this experiment we conclude the following. (a) The **single** index outperforms the **merged**

index, which traverses more links to reach the same neighborhood, thus increasing search latency. This could possibly be because the in- and out-edges of each node in the **merged** index are limited to about $\frac{\ell}{k} = 5\%$ of all points. (b) The **merged** index is still a very good choice for billion-scale k-ANN indexing and serving single-node, easily outperforming the existing state-of-the-art methods and requires no more than 20% extra latency for a target recall when compared to the **single** index. The **single** index, on the other hand, achieves a new state-of-the-art 1-recall@1 of 98.68% with <5 milliseconds latency. The merged index is also a good choice for the DEEP1B dataset. Figure 2(b) shows the recall vs latency curve of the merged $\mathrm{DiskANN}$ index for the DEEP1B dataset built using $k = 40$ shards and $\ell = 2$ on the z840 machine, and with search running on 16 threads.

## 4.4 Comparison on Billion-Scale Datasets: DiskANN **vs IVF-based Methods**

Our final comparisons are with FAISS[18] and IVFOADC+G+P[7], two recent approaches to constructing billion point indices on a single node. Both methods utilize Inverted Indexing and Product Quantization-based compression schemes to develop indices with low-memory footprint that can serve queries with high-throughput and good 1-recall@100. We compare $\mathrm{DiskANN}$ with only IV-FOADC+G+P since [7] demonstrates superior recall for IVFOADC+G+P over FAISS, and moreover, billion-scale indexing using FAISS requires GPUs that might not be available in some platforms.

IVFOADC+G+P uses HNSW as a *routing* layer to obtain a small set of clusters that are further refined using a novel grouping and pruning strategy. Using their open-source code, we build indices with 16 and 32-byte OPQ code-books on the SIFT1B base set. IVFOADC+G+P-16 and IVFOADC+G+P-32 curves in 2(a) represent the two configurations. While IVFOADC+G+P-16 plateaus at 1-recall@1 of 37.04%, the larger IVFOADC+G+P-32 indices reach 1-recall@1 at 62.74%. With the same memory footprint as IVFOADC+G+P-32, $\mathrm{DiskANN}$ saturates at a perfect 1-recall@1 of 100%, while providing 1-recall@1 of above 95% in under 3.5ms. Thus $\mathrm{DiskANN}$, while matching the memory footprint of compression-based methods, can achieve significantly higher recall at the same latency. Compression-based methods provide low recall due to loss of precision from lossy compression of coordinates which results in slightly inaccurate distance calculations.

Zoom[24] is a compression-based method, similar to IVFOADC+G+P, that identifies the approximate nearest $K' > K$ candidates using the compressed vectors, and re-ranks them by fetching the full-precision coordinates from the disk to output the final set of $K$ candidates. However, Zoom suffers from two drawbacks: (a) it fetches all the $K'$ (often close to hundred even if $K = 1$) full-precision vectors using simultaneous random disk reads, which would affect latency and throughput, and (b) it requires expensive $k$-means clustering using hundreds of thousands of centroids to build the HNSW-based routing layer. For example, the clustering step described in [24] utilizes 200K centroids on 10M base set, and might not scale easily to billion-point datasets.

## 5 Conclusion

We presented and evaluated a new graph-based indexing algorithm called $\mathrm{Vamana}$ for ANNS whose indices are comparable to the current state-of-the-art methods for in-memory search in high-recall regimes. In addition, we demonstrated the construction of a high-quality SSD-resident index $\mathrm{DiskANN}$ on a billion point dataset using only 64GB of main memory. We detailed and motivated the algorithmic improvements that enabled us to serve these indices using inexpensive retail-grade SSDs with latencies of few milliseconds. By combining the high-recall, low-latency properties of graph-based methods with the memory efficiency and scalability properties of compression-based methods, we established the new state-of-the-art for indexing and serving billion point datasets.

**Acknowledgements.** We would like to thank Nived Rajaraman and Gopal Srinivasa for several useful discussions during the course of this work.

## Footnotes

[2]The average degree of an NSG index can vary depending on the inherent structure of the dataset, here we assume a degree of 50, which is reasonable for datasets with little inherent structure.

[3]This notion itself was inspired by a related property known as the *Relative Neighborhood Graph* (RNG) property, first defined in the 1960s [16].

[4]Although more complex compression methods like [14, 19, 18] can deliver better quality approximations, we found simple product quantization sufficient for our purposes.

[5]https://github.com/ZJULearning/nsg

[6]Since NSG needs a starting $k$-nearest neighbor graph, we also include the time taken by EFANNA [12].

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
