[Reviews · NeurIPS 2019]

Reviewer 1



The paper under review builds upon prior work on graph-based approximate kNN search, and improves upon by (a) introducing random graph initialization that speeds up the graph construction procedure (b) replacing a hard RNG constraint w/ a soft \alpha-RNG constraint, which reduces the number of hops required by the algorithm (c) expanding the BeamSearch procedure to work with SSD storage, by selecting k nodes to expand at a time, which addresses the I/O bottleneck. Overall, while these all can be thought of as incremental contributions, the resulting algorithm outperforms the relevant baselines. It will no doubt be of interest to practitioners in the field. This type of work, however, could significantly benefit from open-sourcing the code to facilitate adoption. The paper is well written, the techniques are well explained and the connection to prior work is clear, even for non-expert reader. The experimentation is thorough and convincing.

Reviewer 2



This paper studies the problem of process approximate nearest neighbor search with memory and SSD, and propose a graph-based index and search algorithm named Rand-NSG. It can hold a billion points searching on a normal workstation with a cheap SSD. However, there are some concerns: (1). The point of novelty is not strong enough, because the sub-ideas are all existing techniques in other existing researches (e.g. beam search, product quantization), and making previous NSG method SSD-friendly is an incremental work. (2). In Figure 3, it is not enough to only see the distance comparisons between different methods. The runtime of comparison is not the only factor affecting performance. Other overheads should be considered as well. The experiment of "Recall-Queries per second (1/s) tradeoff" should be added, like in https://github.com/erikbern/ann-benchmarks/. Other minor typos: (1). page 2 line 80: "the number the number of random" -> "the number of random"

Reviewer 3



A very nice contribution! The writing could be improved, but it's in general understandable. However, citation quality can be improved. In particular, it seems to me that NSG and HNSW are actually using the same pruning rule (which results in approximate relative neighborhood graph). I really like your updated version, which reduces the number hops (and I haven't seen this pruning variant before)! Detailed comments: Abstract and further: base points sounds like a strange term, do you mean domain points? 22-23 Why 100 dimensions? Not 200-300? 27 You talk about the Euclidean search, but cite a paper on the inner-product search. Please, find a more specific-generic citation that describes this phenomena. Off-the-top of my head *POSSIBLY* more appropriate references: i. Weber, R., Schek, H.J. and Blott, S., 1998, August. A quantitative analysis and performance study for similarity-search methods in high-dimensional spaces. In VLDB (Vol. 98, pp. 194-205). ii. Beyer, Kevin, et al. "When is “nearest neighbor” meaningful?." International conference on database theory. Springer, Berlin, Heidelberg, 1999. 29 Seems a bit awkward. First, it's IMHO better to first define recall. Second, I am not sure we want to maximize recall, as the maximum recall is 100% and as you mention it's not a realistic target. What we really want to maximize efficiency at a given recall level (likely somewhat close to 100%, albeit the desired level of recall might be application dependent). 37 This is a controversial claim! See, e.g., results of ann-benchmarks. There are bunch of methods whose performance is roughly the same. See the latest comparison showing that graphs based on NN-descent algorithms work really well too (about the same performance as HNSW): Li, Wen, et al. "Approximate nearest neighbor search on high dimensional data-experiments, analyses, and improvement." IEEE Transactions on Knowledge and Data Engineering(2019). Crucially: are these methods all that different? HNSW and NSG are using basically the same graph pruning heuristic (approximation to the relative neighborhood graph). It seems that you fail to disclose this. 39 What is exactly a navigable graph? 40 IMHO, the procedure is not a *GREEDY* walk. It's a best-first graph traversal with a buffer priority queue that keeps a buffer list of candidates. 53-54 awkward writing 127. the main idea in [12]. Actually, this idea has been used in a bunch of other papers, notably in HNSW and its predecessor NSW. 154-156 Although implementationally different, your algorithm is conceptually similar to the NN-descent implemented in Wei Dong's k-graph. https://github.com/aaalgo/kgraph 196. The merging algorithm isn't quite new please cite the relevant paper: i. I known this approach from Scalable k-nn graph construction for visual descriptors J Wang, J Wang, G Zeng, Z Tu… - 2012 IEEE Conference …, 2012 - ieeexplore.ieee.org ii. However, Wang et al. learned it from J. L. Bentley. Multidimensional divide-and-conquer. Commun. ACM, 23(4):214–229, 1980. 1, 2 249 Which public benchmarks?

[Author Response · NeurIPS 2019]

# Rebuttal for submission 7667 - Rand-NSG

**Reviewer 1:** We will release our code along with the paper.

**Reviewer 2:**
Our main goal was to enable high performance SSD-friendly indices on inexpensive workstations. Towards this goal, as you have suggested, we have adapted known techniques, and, where necessary, contributed new techniques as well. A few contributions are: (1) We introduce a new pruning strategy parameterised by $\alpha$ that reduces the number of hops from navigating vertex to any other vertex. This provides a greater control over the diameter of the graph which is important for disk search. This is a feature that previous methods like NSG and HNSW lack. Using our pruning with a higher $\alpha > 1$ parameter will yield more longe range edges and reduces the diameter of the graph. The practical benefits of this idea can be seen in Figure 2(b) in the submission, where we notice that the number of hops required for search improved with the degree of the graph for Rand-NSG, and plateaus for NSG and HNSW. (2) We also introduce a 2-pass construction algorithm, that allows us to use lower $L$ for index construction, thus improving the construction speed necessary for a target graph index quality (compared to NSG and HNSW). (3) We provide a high quality implementation of disk-based search.

**QPS plots:** We will expand on the plots above comparing NSG, HNSW, and Rand-NSG. Rand-NSG indices can be searched with NSG in-memory search (we did not implement a separate in-memory search since our focus was on disk search). The plots above suggest that Rand-NSG indices are competitive with NSG and HNSW. Considering that they use roughly the same search algorithm for querying, we focused on the number of distance comparison per query as a machine-agnostic way of comparing the algorithms.

**Reviewer 3:** Thank you for pointing out citations we have missed or erred on (e.g., in lines 27, 127, 154, 196, 249). We will update related work, give due credit to missed citations and do a pass over the writing. Re. other questions:

- Yes, base points mean domain points. We'll clarify the notation.

22-23 : Why not $200 - 300$? Could well be, we have experimented with 960-dimensional GIST in the paper.

29 : We will define the recall formally in the final version. *a-recall@b* means the percentage of points among the $a$ points that intersect with the true $b$ nearest neighbor. We'll also clarify that the goal is maximizing search efficiency for a given recall, rather than the recall itself.

37 : We will temper the claim and suggest that graph-based methods produce some of the best indices. We'll give NN-descent algorithms credit in related work.

39 : We do not have a concrete definition of *navigable graphs*; we roughly mean those graphs on which greedy-like iterative search heuristics converge rapidly.

40 : We will clarify the differences between our search and pure greedy search.

- We will report numbers on Deep1B. We have preliminary evidence suggesting that we can achieve 95% recall@1 with 5ms latency and 30GB working memory.

We'll also discuss the pruning strategy used by HNSW and NSG in our final version, and contrast with Rand-NSG.



[Meta-Review · NeurIPS 2019]

there were some complaints about the novelty of the methods. In post rebuttal discussions, reviewers concurred in subsequent discussions that the paper presents solid state of art implementation and very impressive results, which will have good impact for practitioners. This significant impact by itself was worthy of publication.